# Microphysical and Compositional Differences Between Saharan and Middle Eastern Dust Revealed by UAS Observations

Maria Kezoudi<sup>1</sup>, Alkistis Papetta<sup>1</sup>, Konrad Kandler<sup>2</sup>, Claire L. Ryder<sup>3</sup>, Andreas Leonidou<sup>1</sup>, Christos Keleshis<sup>1,4</sup>, Chris Stopford<sup>5</sup>, Troy Thornberry<sup>6</sup>, Rodanthi-Elisavet Mamouri<sup>7,8</sup>, Jean Sciare<sup>1</sup>, and Franco Marenco<sup>1</sup>

Correspondence: Maria Kezoudi (m.kezoudi@cyi.ac.cy)

Abstract. The rising frequency of mineral dust events in the eastern Mediterranean underscores the need for high-resolution observations to better characterize their properties and impacts. This study reports results from the Cyprus Fall Campaign 2021, which aimed to test and validate a new cost-effective methodology for quantitative dust measurements using GPAC, POPS, and UCASS sensors on-board Uncrewed Aerial Systems(UAS). The Cyprus Fall Campaign 2021 captured the microphysical characteristics of dust particles from two major global sources: North Africa(NA) and the Middle East(ME). The campaign took place between 18/10/2021 and 18/11/2021 with continuous ground-based remote-sensing measurements, complementing 36 UAS flights. This work represents the first intensive UAS-based dust characterization campaign in Cyprus and the wider Mediterranean region during the autumn season. Integrated remote-sensing, in-situ, and trajectory analyses revealed NA dust heights up to 7km over Cyprus, compared to 3.8km for ME dust. Impactor sampling demonstrated a near-1 collection efficiency for particles between 4-14 μm, highlighting its effectiveness onboard the UAS. Particle volume size distributions showed a fine-mode peak at 0.25 μm in both cases, and distinct coarse-mode peaks at 2.2 μm and 4.8 μm for NA and ME dust, respectively. High-altitude impactor samples showed two distinct dust signatures: NA dust enriched in kaolinite-like and Ca-bearing phases, and ME dust dominated by illite/muscovite and Fe-rich components, indicating contrasting source characteristics influenced by granulometry, transport, and atmospheric processing. This study showcases the capability of high-resolution UAS sampling to characterize atmospheric dust and improve understanding of its regional and climatic impacts.

<sup>&</sup>lt;sup>1</sup>CARE-C, The Cyprus Institute, Nicosia, Cyprus

<sup>&</sup>lt;sup>2</sup>Institute of Applied Geosciences, Technical University of Darmstadt, Darmstadt, Germany

<sup>&</sup>lt;sup>3</sup>Department of Meteorology, University of Reading, Reading, UK

<sup>&</sup>lt;sup>4</sup>Innovation Centre, Cyprus Marine and Maritime Institute, Larnaca, Cyprus

<sup>&</sup>lt;sup>5</sup>University of Hertfordshire, Hatfield, United Kingdom

<sup>&</sup>lt;sup>6</sup>National Oceanic and Atmospheric Administration (NOAA), United States

<sup>&</sup>lt;sup>7</sup>ERATOSTHENES Centre of Excellence, Limassol, Cyprus

<sup>&</sup>lt;sup>8</sup>Cyprus University of Technology, Limassol, Cyprus

#### 1 Introduction

Mineral dust is a key constituent of the Earth's system, affecting the radiative balance, cloud properties, and precipitation (Teller et al., 2012; Boucher et al., 2013; Kok et al., 2017). It also has an effect on oceanic and terrestrial biogeochemical processes and atmospheric chemistry. Dust aerosols are mobilised by saltation and sandblasting of soil grains. It can be transported over thousands of kilometres throughout the free troposphere (Ginoux et al., 2001; Kok et al., 2012; Ginoux et al., 2012; Engelstaedter et al., 2006). To understand and quantify the impact of mineral dust on the Earth system, more information is required on height-resolved size distributions, number concentrations, and composition of airborne mineral dust particles (Formenti et al., 2011; Ryder et al., 2013; Weinzierl et al., 2009).

The Eastern Mediterranean basin, due to its proximity to the arid areas of North Africa (NA) and the Middle East (ME), experiences frequent dust episodes throughout the year (Kaskaoutis et al., 2019). The total incidence of days of dust in the Eastern Mediterranean has exhibited a significant upward trend, with the average rate increasing by 2.7 days per decade (Ganor et al., 2010). Dust activity in the region is heavily influenced by weather conditions and climate perturbations (Zittis et al., 2022; Shaheen et al., 2021; Hoerling et al., 2012; Achilleos et al., 2020). The main cyclogenesis zones in the Mediterranean basin are characterized by dust uplift and transport throughout the region (Alpert et al., 1990; Karam et al., 2010). Correlation maps associate dust-rich years with high cyclonic activity in the Mediterranean (Dayan et al., 2008). Intense dust episodes over the Eastern Mediterranean are typically associated with phenomena such as the Cyprus Low (Kalkstein et al., 2020; Dayan et al., 2008) and the Sharav cyclones (Karam et al., 2010). These phenomena, transport dust from the Arabian and northern Sahara deserts to the Eastern Mediterranean basin (Nisantzi et al., 2015; Mamouri et al., 2016).

Cyprus, located at the crossroads of the transport pathways, is affected by an increasing number of dust episodes (Achilleos et al., 2014, 2020). Dust transport episodes exhibit significant seasonal and yearly variations, with spring highlighted as the peak period over Cyprus (Kallos et al., 2006; Camps et al., 2015; Papadimas et al., 2017). Cyprus continues to experience dust events in autumn, primarily from ME air masses (Nisantzi et al., 2015; Achilleos et al., 2020). Given that fall dust events were comparatively less studied than those in other seasons (Kallos et al., 2014; Ganor et al., 2010), the present work aims to characterize the dust transported during autumn from different source regions, with particular emphasis on the vertical distribution and the microphysics of the layers, including vertically-resolved particle size distribution and mineralogy.

Height-resolved observations of mineral dust particles can be conducted using remote-sensing instrumentation (Mona et al., 2012; Sugimoto and Zhongwei, 2014; Toledano et al., 2019; Mamouri and Ansmann, 2015) as well as airborne means, such as sensors deployed on-board balloons (Kezoudi et al., 2021b), aircraft (Liu et al., 2018; Weinzierl et al., 2017; Ryder et al., 2015; Marenco et al., 2018) and Uncrewed Aerial Systems (UAS) (Mamali et al., 2018; Kezoudi et al., 2021a; Rohi et al., 2020). UAS offer cost-effective vertically-resolved in-situ atmospheric observations within the lower troposphere, complementary to ground-based remote-sensing and in-situ observations. UAS are also well suited for measuring dust in areas inaccessible to crewed aircraft (e.g. during volcanic eruptions or other extreme events), with the added advantage of safe and easy payload recovery post-flight (Thomas et al., 2018; Zampolli et al., 2019; Lee et al., 2020).

60

75

80

The Unmanned Systems Research Laboratory (USRL; https://usrl.cyi.ac.cy) of the Cyprus Institute, part of the EU Aerosol, Clouds and Trace Gases Research Infrastructure (ACTRIS), provides mobile UAS-sensor solutions for deployment across Europe and beyond. Through a transnational access scheme, it supports research, innovation, and training within the European Union Aviation Safety Agency's (EASA) drone regulations. The USRL airfield in Orounda is strategically located 6.5 km from the Cyprus Atmospheric Observatory at Agia Marina Xyliatou (CAO-AMX; https://cao.cyi.ac.cy/agia-marina-xyliatou). CAO-AMX provides air pollutant observations using numerous ground-based in-situ and remote-sensing instruments. It serves as the reference rural background station for the national air quality network and is part of the European networks ACTRIS and the European Monitoring and Evaluation Programme (EMEP). Another CAO station with similar instrumentation is located at the Cyprus Institute premises (CAO-Nicosia), at a distance of 28 km from USRL airfield, and is classified as an urban background station.

The data presented in this study were collected during the Cyprus Fall Campaign 2021, which had two primary scientific objectives: (1) to investigate the microphysical properties of airborne dust particles from different source regions over Cyprus during autumn period, and (2) to establish a robust methodology combining the relevant airborne and remote-sensing instruments to optimize dust sampling and characterization. This campaign spanned one month, running from 18th of October to 18th of November 2021. Throughout the campaign, measurements were continuously collected by ground-based remote-sensing instrumentation at CAO-AMX and CAO-Nicosia, complementing the scientific UAS flights. A total of 36 UAS flights were conducted during two distinct week-long dust events. During most of the days when UAV flights were performed, the daily-averaged Aerosol Optical Depth (AOD) exceeded 0.2 over Nicosia. This paper represents the first study in Cyprus and the broader Mediterranean region dedicated to the results of an intensive campaign using UAS-based sensors amidst dust episodes occurring during the fall season. The structure of this paper is as follows: Section 2 provides an overview of the instrumentation and methods employed. The findings are elaborated on and discussed in Section 3. Concluding remarks and summary are presented in Section 4.

## 2 Observations and Methods

## 2.1 Campaign Overview

The Cyprus Fall Campaign 2021 aimed to study the microphysical properties of mineral dust transported to Cyprus using airborne and ground-based observations. An intensive UAS campaign was conducted at the USRL private runway in Orounda from 18th of October to 18th of November 2021. The UAS were fitted with a suite of instruments dedicated to in-situ particle measurements, comprising two Optical Particle Counters (OPCs), two Compact Optical Backscatter Aerosol Detectors (COBALDs), and impactors designed for particle collection within dust layers. Continuous ground-based remote-sensing measurements, incorporating lidars, ceilometers, and sun-photometers, were performed at the CAO-AMX and CAO-Nicosia atmospheric stations and at the Cyprus Atmospheric Remote Sensing ACTRIS National Facility (CARO-LIM NF) of the ERATOS-THENES Centre of Excellent in Limassol.

To identify dust events and effectively plan UAS flight operations, we utilised forecasts from several dust models, i.e. CAMS and SKIRON (Nickovic et al., 2001; Inness et al., 2019), complemented by real-time observations from remote-sensing instruments. Two distinct dust events were identified, the first occurring between 25th of October and 1st of November, with daily-averaged AODs reaching up to 0.3 and air mass originated from North Sahara as determined by the HYSPLIT model. The second event occurred between 13th and 18th of November and the primary source of air masses was traced back to the ME or Levant region. Height-resolved information for each dust event was captured from its onset to dissipation using daily UAS-based OPCs and continuous ground-based lidar observations. These instruments provided detailed information on the evolution of aerosol properties throughout the atmospheric column.

Table 1 presents an overview of the UAS flights conducted during the campaign, detailing the conditions and observed dust layers as reported by the full suite of airborne and ground-based instruments deployed. This includes the take-off time (ToT) of the UAS, the AOD during the flight from the AMX sun-photometer. In addition, it provides information on the elevated dust layers captured by the UAS and their source region retrieved from back trajectories over Orounda. The predominant origin of air masses during the dusty days is distinguished between the two deserts, NA and ME.

**Table 1.** Details on the conditions during the UAS flights of the Cyprus Fall Campaign 2021, including date (green colour: no dust observed with UASs, orange colour: dust observed with UASs), UAS Take-off Time (ToT), AOD during the flight from the sun-photometer in AMX, altitude range of dust layer from lidar, Boundary Layer (BL) height from the ceilometer in AMX, UAS flight ceiling altitude, dust elevated layer from the UAS flight, the source region of the observed UAS elevated dust layers. P stands for POPS and U for UCASS. All altitudes are given in kilometers ASL.

| Date       | UAS ToT | AOD    | Elevated dust | BL height | UAS     | Elevated dust | Source Region |  |
|------------|---------|--------|---------------|-----------|---------|---------------|---------------|--|
| Date       | (UTC)   | 500-nm | layers lidar  |           | ceiling | layers UAS    | HYSPLIT       |  |
| 18/10/2021 | P:1330  | 0.1    | No dust       | n/a       | 2.8     | /             | No dust       |  |
|            | U:1430  | 0.1    |               |           | 3.4     | n/a           |               |  |
| 24/10/2021 | P:0900  | 0.13   | 5-7           | 0.8       | 4.0     | n/a           |               |  |
|            | U:1100  |        |               | 0.6       | 3.6     | 11/a          |               |  |
| 25/10/2021 | P:1300  | 0.28   | 3-4.7         | 1.6       | 3.6     | 3.0-3.6       |               |  |
| 23/10/2021 | U:1430  | 0.20   | 2.5-5.2       | 1.6       | 4.1     | 2.5-4.1       |               |  |
| 27/10/2021 | U: 1030 | 0.3    | 2.1-4.4       | 0.9       | 3.9     | 2.0-3.5       | AFRICA        |  |
|            | P: 1145 | 0.5    |               | 0.9       | 3.8     |               |               |  |
| 28/10/2021 | P: 0815 | n/a    | 1.0-2.7       | 0.8       | 3.5     | 1.0-2.7       |               |  |
|            | U: 0930 | 11/4   |               | 0.8       | 3.8     |               |               |  |
| 29/10/2021 | P:0830  | 0.18   | 2.0-3.2       | 1.3       | 2.8     | 2.0-2.8       |               |  |
|            | U:0700  |        | 2.0-2.8       | 0.8       | 3.4     |               |               |  |
| 31/10/2021 | P:1230  | n/a    | 2.6-3.2       | 0.9       | 3.0     | 2.6-3.0       |               |  |
| 31/10/2021 | U:1330  |        | 1.7-3.5       | 1.1       | 3.7     | 1.7-3.5       |               |  |
| 04/11/2021 | P:1300  | 0.06   | No dust       | 1.5       | 3.1     | n/a           | No dust       |  |
|            | U:1430  |        |               | 1.7       | 3.5     |               |               |  |
| 13/11/2021 | P:1300  | 0.2    | 1.4-2.4       | 1.3       | 3.6     | 1.4-2.4       |               |  |
|            | U:1430  |        |               | 1.3       | 3.9     |               |               |  |
| 14/11/2021 | U:1230  | 0.26   | 1.4-3.5       | 1.0       | 4.0     | 1.4-3.5       |               |  |
|            | P:1400  |        | 1.4-3.4       | 1.0       | 3.1     | 1.4-3.1       |               |  |
| 15/11/2021 | U: 1230 | 0.28   | 1.4-3.4       | 1.3       | 4.3     | 1.3-2.8       | M.EAST        |  |
|            | P:1345  |        |               | 2.1       | 3.2     |               |               |  |
| 16/11/2021 | U:1315  | 0.3    | 1.4-2.9       | 2.2       | 4.8     | 1.4-2.9       |               |  |
| 17/11/2021 | P:1300  | n/a    | 1.4-3.5       | 2.8       | 3.1     | 1.4-3.1       |               |  |
| 1//11/2021 | U:1730  | 11/a   | 1.4-3.8       | 2.4       | 4.6     | 1.4-3.8       |               |  |
| 18/11/2021 | P:1300  | 0.35   | 1.4-3.4       | 1.7       | 3.2     | 1.4-3.1       |               |  |
| 16/11/2021 | U:1430  | 0.55   | 1.4-3.1       | 2.2       | 4.5     | 1.4-3.1       |               |  |

# 95 2.2 The UAS used for this campaign

This study utilized two UAS models, the I-Soar and Skywalker, as presented by Kezoudi et al. (2021a). Both UAS belong to the category of fixed-wing aircraft and are mainly constructed with foam plywood. Equipped with a datalogging system, and GPS, these UASs were remotely operated via a Ground Control Station (GCS). The UAS were equipped with meteorological sensors and a pitot-tube, which continuously collect data at a rate of 1 Hz. These sensors provided measurements of Relative Humidity, temperature, air pressure and airspeed, enriching the dataset and facilitating a comprehensive understanding of atmospheric conditions. The duration of UAS flights in this study typically ranged from 50 to 80 minutes, constrained primarily by battery capacity.

#### 2.3 Instruments on-board UASs

## 2.3.1 Optical Particle Counters

Two OPCs were deployed on-board the two different UAS for this experiment, with approximately one hour interval between their take-off times. The Portable Optical Particle Spectrometer (POPS; et al. 2016), is a small light-weight and high sensitivity OPC, which was deployed on-board the I-Soar UAS (Kezoudi et al., 2021a). The instrument operates a 405-nm laser diode, and a calibrated Mie theory calculation is used to determine the particle size based on the intensity of scattered light. POPS collects light scattered by individual particles in an angular range between 38° and 142°. The diameter of the inlet tube is 1 mm and the sample flow rate is around 3 cms<sup>-1</sup>, yielding a flow velocity of 3.8 ms<sup>-1</sup>. POPS is able to accurately measure particles with diameters ranging from 0.1 to 3.4μm. The instrument was calibrated by the manufacturer using latex spheres with a refractive index of 1.615 – 0.001*i*.

To extend the size range of measured particles towards coarser sizes, the Universal Cloud and Aerosol Sounding System (UCASS; Smith et al. 2019) was integrated below the wings of the Skywalker 2015 UAS (Kezoudi et al., 2021a). The UCASS is a lightweight OPC that was designed for use as a balloon-borne instrument, as a dropsonde, or on-board UAS (Girdwood et al., 2022)). The geometry of the instrument minimizes particle losses and droplet shattering at the inlet. It uses a 658 nm laser diode to detect light scattered by particles between  $16^{\circ}$  and  $104^{\circ}$ . Computational Fluid Dynamics (CFDs) simulations confirm that integrating the UCASS beneath the wings enables its airflow measurements to align with those obtained from the UAS's nose-mounted pitot tube (Girdwood et al., 2022). The UCASS units were calibrated for dust particles by the manufacturer, using a refractive index of 1.52 + 0.002i (typical of dust) and a particle diameter range of  $0.5-21.0 \, \mu m$ .

Particle size distributions in this study were calculated using the formulas provided in Kezoudi et al. (2021b). Specifically, raw particle counts C for each size bin i were used to calculate the particle number concentration per size bin  $n_i = C_i/v$ , per unit volume (v), per second over the covered size range. Summing  $n_i$  across all size bins yields the total number concentration N. Assuming spherical particles, the particle number (1) and volume (2) size distributions are calculated using the sum of the number and volume concentration, respectively, for each bin together with the logarithmic bin centre  $log D_{c,i} = (log D_{i+1} + log D_i)/2$  and width  $dlog D_i = log D_{i+1} - log D_i$ , where  $D_i$  are the bin edge diameters, by

$$\mathrm{d}n_i/\mathrm{dlog}D_i = \frac{n_i}{\mathrm{dlog}D_i} \tag{1}$$

$$\mathrm{d}V_i/\mathrm{dlog}D_i = \frac{\pi n_i}{6} \frac{D_{c,i}^3}{\mathrm{dlog}D_i}. \tag{2}$$

To achieve a complete particle size distribution for both POPS and UCASS, the size bins of the two instruments were combined. For the size range between 0.1 and 2.3 µm, POPS data were solely utilized in the analysis. For particle sizes exceeding 2.3 µm, UCASS data were employed. This cutoff diameter was chosen based on prior research that indicated possible artifacts of size

150

155

around 2.2,µm of ambient measurements within this size range when using UCASS (Kezoudi, 2020). Figure 1 shows the combined volume size distributions averaged over the observed dust layers, as obtained with POPS and UCASS UAS flights on 31st of October and 15th of November 2021. Figure 1a includes the size distributions in the full size range of the two instruments, which demonstrates good resemblance in the overlap size section of POPS and UCASS. Figure 1b depicts the volume size distribution in the combined size range of the two OPCs, as described above. The error bars shown in the figure represent the standard deviation of the measurements over the whole flight, indicating the variability of the data collected within the dust layers.

Estimation of aerosol mass concentration relies on the assumption that the particles are spherical, allowing simplified volumetric calculations to be applied. Given an assumed particle density  $(\rho)$  and an assumed size distribution, the mass concentration (M) can be derived from the particle volume concentration (V) as:

$$\mathbf{M} = V \cdot \rho \tag{3}$$

where V is obtained from the particle size distribution using an appropriate integration approach (i.e., a lognormal distribution).

This approach is commonly used in remote sensing and in-situ aerosol studies to convert optical or number concentration data into mass concentration estimates (Dubovik et al., 2002; Kahn et al., 2005)).

A representative density of  $1.6 \ g/cm^3$  was adopted for fine/submicron aerosols for mass-from-volume conversions, consistent with commonly used methodological assumptions and measured densities reported in the literature (McMurry et al., 2002; Zhao et al., 2017; Saide et al., 2020). In contrast, coarse-mode aerosols, particularly those dominated by mineral dust from sources such as the Sahara, exhibit higher densities around  $2.6 \ g/cm^3$  (Maring et al., 2003). This density assumption aligns with the compositional analyses of Saharan dust, which consists mainly of silicates, iron oxides, and other mineral components (Formenti et al., 2011). These density values are widely used in climate modeling and atmospheric aerosol retrieval algorithms, particularly in satellite-based assessments and aerosol transport simulations (Koffi et al. (2016); Kok et al. (2017)). However, variations in aerosol composition and mixing state can introduce uncertainties in mass concentration estimates, necessitating further validation through in-situ measurements. In this study, the fine mode was defined as the size range from 0.1 to 0.7  $\mu$ m with particle density of  $1.6 \ g/cm^3$ . For particles larger than  $0.7 \ \mu$ m, a density of  $2.6 \ g/cm^3$  was used. The threshold of  $0.7 \ \mu$ m was chosen based on the measurements, as it corresponds to the minimum between the fine and coarse modes, observed during most of the flights.

## 2.3.2 Particle sampling with the Giant Particle Collector

A miniaturized and 3D-printed version of the Giant Particle Collector (GPaC; Kezoudi et al. 2021a), a body impactor shown in Figure 2, has been designed for UAS applications (Lieke et al., 2011). In principle, there is no upper cut-off diameter for dust particle collisions with the adhesive sampling substrate as it moves through the air. However, the upper limit is mainly limited by the sticking efficiency of the substrate. In this manuscript, the investigated size range is up to 21 µm particle diameter, as

**Figure 1.** Volume size distribution of (a) POPS (square points) and UCASS (triangle points) as derived for the UAS measurements on 31th of October (red) and 15th of November 2021 (blue); and (b) combined POPS and UCASS for the same cases. The vertical dashed line shows the size bin where UCASS data are merged with POPS. For clarity only upper error bounds are shown.

**Figure 2.** Design of the GPAC in the open configuration. Following sampling, the substrate holder is retracted into its protective cover to prevent contamination.

has been observed with the UCASS. In our case, particles down to approximately  $1\,\mu\mathrm{m}$  diameter could be sampled with the system.

Two GPaC samplers (Figure 2) were integrated beneath the wings of the Skywalker 2015 UAS, alongside the UCASS units, to facilitate airborne dust collection. Each sampler holds a pure carbon adhesive substrate (12.5 mm diameter, SpectroTab, Plano GmbH, Wetzlar, Germany) mounted on a standard Single-particle Electron Microscopy (SEM) aluminium stub at the

170

tip of the GPaC system. Before each flight, the substrates were mounted, and during flight, the pilot could manually expose a substrate to the airstream at a predetermined altitude and for a specified duration. After sampling, the holder retracted the substrate back into its protective cover, and the same procedure was repeated for the second substrate to sample a second atmospheric layer. At the end of the flight, both samples were dismounted and stored for offline analysis.

For SEM, optimal particle spacing is crucial; there must be enough particles for statistical relevance, but not so many that they overlap on the substrate (Kandler et al., 2018). A typical ideal spacing on the substrate is 20–50 µm. Since particle collection depends on both the length of the sampling path and the aerosol concentration, which is unknown before flight, the length of the path must be planned based on typical or expected conditions. If the actual concentration during flight is much lower or higher than estimated, the collected sample may not be suitable for analysis. As a result, only 16 of the 22 collected samples had suitable coverage for analysis; the remaining six, due to low loading or contamination introduced during sample handling, are excluded from further discussion.

Particle concentrations from the GPAC were determined based on the particle number per sample, sampling time, and UAS airspeed. The total concentration for each sample was obtained by summing all values corresponding to the individual particle.

Figure 3 shows the collection efficiency of particles collected by the GPaC relative to those measured by the UCASS during the campaign. The apparent collection efficiency for particle size at around 1  $\mu$ m was found to be 0.1. A notable decrease in efficiency occured at approximately 2.5  $\mu$ m, corresponding to a slight artifact in the UCASS, as discussed in Section 2. The average collection efficiency increased to nearly 1 for particles sized between 4 and 14  $\mu$ m. The efficiency decreased again to about 50% for particles larger than 15, $\mu$ m. The figure demonstrates the generally comparable values between the GPaC and the UCASS, indicating suitable performance of the methodology across the particle size range.

**Figure 3.** The apparent collection efficiency of particles collected by GPaC relative to those measured by UCASS during dust events across the campaign period. Solid line represents the mean values.

## 2.3.3 Single particle electron microscopy

Automated SEM analysis was performed on the collected samples using a FEI Quanta400F microscope (FEI, Eindhoven, The Netherlands). Non-carbonaceous particles were identified by their brighter backscatter electron signal relative to the carbon substrate and segmented from the background using image analysis (Kandler et al., 2018). For the present work, a total of 16,200 particles were analyzed. The particle sizes were calculated from the projected area visible in the electron microscope with a set of corrections outlined in Kandler et al. 2018.

Based on chemical quantification, particles were classified into compositional groups. This study adopts the classification framework of Kandler et al. (2020), with an adaptation in the classification limits to ease an over-representation of kaolinite-like particles. All data shown are reprocessed with the new scheme (see Appendix A1). In addition, a simplified scheme was employed to distinguish dust from non-dust particles, using the combined concentrations of Ca, Si, Al, Ti, and Fe versus those of Na, Cl, and S. The specific classification criteria are provided in Table A1 in the appendix. It should be noted that SEM with Energy-Dispersive X-ray analysis yields elemental compositions rather than definitive mineralogical identifications; therefore, classified particles are referred to as "mineral-like."

The central 95% confidence intervals for the average sample composition and the number of particles per class were estimated by bootstrapping (DiCiccio and Efron, 1996; Virtanen et al., 2020). Data from previous campaigns used for comparison are available in public data repositories: Morocco (Panta et al., 2023), Tenerife (Waza et al., 2019), Barbados (Kandler et al., 2018), Tajikistan (Kandler et al., 2019a) and Svalbard (Kandler et al., 2019b).

#### 2.4 Ground-based Remote-Sensing instruments

#### **2.4.1** Lidars

A CE376 dual-wavelength polarization lidar (CIMEL, France) was continuously operated at CAO-Nicosia during the campaign. This compact system uses laser diode and Nd:YAG technology, emitting at 808-nm and 532-nm with a 4.7 kHz repetition rate. It features one reception channel for the infrared and two for green (co-polar and cross-polar). It records height signals every 15 m from 0.1 km to 30 km, with a 1-sec integration time. Raw CIMEL lidar data undergo pre-processing to correct detection errors and eliminate ambient background noise across all three channels before being used for depolarization characterization, as described in Papetta et al. (2024).

A second lidar system, the multi-wavelength PollyXT, was also continuously operated at the CARO-LIM NF site during the campaign. This advanced lidar enables high-resolution profiling of aerosol backscatter, extinction, and depolarization properties, providing comprehensive insight into aerosol optical characteristics throughout the atmospheric column (Mamouri et al., 2023).

### 2.4.2 Ceilometer

The Vaisala Ceilometer CL51 is a fully automated lidar system designed for continuous operation in all weather conditions Münkel et al. (2007). Utilising pulsed diode laser technology, it emits short powerful laser pulses vertically or near-vertically to report attenuated backscatter profiles. It covers a vertical range of up to 15 km. By analysing backscatter caused by clouds, precipitation and aerosols, it accurately determines cloud base and boundary layer height. Two CL51 units were operational during the campaign, one at CAO-Nicosia, and another one at the CAO-AMX.

#### 2.4.3 Sun-photometers

During the campaign, three sun and sky scanning spectral radiometers from the AErosol RObotic NEt- work (AERONET; Holben et al.,1998) were employed for measurements. A lunar/sun- sky- photometer CE318T was at CAO-Nicosia, and two CE318 was at CAO-AMX and at CARO-LIM NF. The CIMEL model used at both locations is capable of direct-sun and diffuse sky measurements across eight spectral bands ranging from 340-nm to 1020-nm. The output of the AERONET network includes aerosol property parameters such as Aerosol Optical Depth (AOD) and Angstrom exponent, alongside other properties derived from inversions of sky radiance observations.

**Figure 4.** HYSPLIT endpoints for sensitivity study of back-trajectories for dates 31/10/2021 13:00 UTC (arriving at 2.5 km a.g.l.) and 15/11/2021 13:00 UTC (arriving at 2 km a.g.l.). Multiple trajectories are initiated from the selected ending location, calculated by offsetting meteorological data with a fixed grid factor.

## 2.5 Back-trajectories Model

The Hybrid Single-Particle Lagrangian Integrated Trajectory model (HYSPLIT; Stein et al., 2015; Rolph et al., 2017) was run with Global Data Assimilation System (GDAS) meteorological reanalysis fields at approximately 50 km resolution to investigate the origins of observed airmasses over Cyprus. For each UAS flight, we specifically computed 5-day backward trajectories ending at the Orounda airfield. These trajectories were calculated for arrival heights ranging between 1.0 and 5.0 km Above Sea Level (ASL), depending on the height of the dust layer for each case. This approach allowed us to infer the atmospheric pathways leading to the study area, providing insights into the source regions and movement of air masses during the campaign period.

To assess the sensitivity of the backward trajectories, we used the grid ensemble approach see e.g. Marenco et al. 2006 within HYSPLIT. Trajectories are automatically computed around a 3-dimensional cube centred around the ending point. The 3-dimensional cube consists of 27 points on 3 planes of 9 trajectories, each plane at  $\pm 250$  m altitude. The 9 trajectories on each plane have a horizontal grid spacing of 1° latitude by 1° longitude (120 km). Notably, the final positions of the trajectories are left constant, with only the meteorological data points associated with each trajectory being offset. This ensures that all trajectories originate from the same geographical point. Figure 4 illustrates two examples of this sensitivity analysis.

#### 3 Results and Discussion

# 245 3.1 Remote Sensing Observations

Figure 5 shows the daily averaged AODs (Level-2) measured by the sun-photometers at the CAO-Nicosia and CAO-AMX stations, alongside the corresponding UAS flying days, indicated with the icon of an aircraft. A gradual increase in AOD is observed from 24th of October showing the arrival of the dust plume. The daily averaged AODs over AMX and Nicosia remained relatively stable, staying above 0.15 from 24th of October to 1st of November. AODs at AMX decreased to below 0.1

from 2nd to 10th of November, while slightly larger values were observed by the Nicosia sun-photometer during this period, which can be explained by local sources. A gradual increase in AODs is noted at both stations from 11th of November onwards. The highest values of daily-averaged AODs, approximately 0.42 and 0.40 were recorded on 17th and 19th of November over Nicosia, and 0.35 over AMX.

**Figure 5.** Overview of daily-averaged AOD at Agia Marina Xyliatou (AMX) and Nicosia during the 1-month campaign period from 18th of October to 18th of November 2021, obtained from AERONET sun-photometer observations. Flying days are indicated with the image of an airplane. Yellow-shaded area indicates the dust event period where UAS flights were performed. Dashed lines indicate absence of data for a specific day.

Figure 6 presents a comprehensive overview of the vertical structure of aerosol layers over Nicosia from the lidar measurements between 18th of October and 19th of November 2021. This is derived from the height-resolved observations of the
Range Corrected Signal (RCS) in the green channel and volume depolarization ratio recorded by the Cimel aerosol lidar. The
first days of the campaign were characterised by minimal dust presence, as evidenced by the observed Volume Depolarization
Ratio (VDR) levels consistently below 0.05. This suggests that the increased backscatter signal within a range of 3 km ASL
from 18th to 25th of October is likely attributed to local pollution within the boundary layer. Similar atmospheric conditions
were observed by the PollyXT lidar in Limassol (Figure 7).

The first dust intrusion, classified as moderate, persisted for about one week between 26th of October and 2nd of November, observed simultaneously at both sites. Thin layers of dust were detected by both lidars on 25th and 26th of October, coinciding with an increase in the VDR to 0.2 in the height layer between 2.5 and 4 km ASL. By 27th and 28th of October, the depolarising layer extended from surface to 4.2 km ASL, as shown in Figures 6b and Figures 7b. The VDR peaked at 0.2 on the last day of the event (2nd of November) at 2.3 km ASL. Starting from 2nd of November, dust-free conditions prevailed again, with

observed VDR levels dropping below 0.05, which is an indication that the backscatter signal up to 2 km is associated with local pollution (Illingworth et al., 2014).

The second dust event started with faint traces on 13th of November 2021, followed by the main plume's arrival between 1 and 3 km altitude ASL on 14th of November. The lidar shows a homogeneous dust layer in the volume depolarisation ratio, fluctuating between 0.15 and 0.2 from 13th to 18th of November and with variable layer top between 2 and 4 km. This event was characterised by greater homogeneity in the vertical structure and higher VDR and AOD compared to the first dust event.

**Figure 6.** Time evolution of the range-corrected signal of the 532-nm Total (a) and volume depolarization ratio (b) from the lidar in Nicosia during the Fall Campaign period. Profiles are shown above the overlap region of the lidar. Clouds have a large RCS and therefore shown in grey-white colour in panel a. A gap on the data is seen between 14:00 UTC on 26th of October and 05:00 UTC on 27th of October due to technical issues.

**Figure 7.** Time evolution of the attenuated backscatter of the 532nm Total (a) and volume depolarization ratio (b) from the PollyXT-CYP lidar of CARO-LIM NF in Limassol, during the Fall Campaign period. Profiles are shown above the overlap region of the lidar. A gap on the data is seen between 3rd and 4th, and 7th and 9th November 2021 due to technical issues.

#### 3.2 Airborne observations - OPCs

275

280

During the first dust event, the initial traces of the dust plume were detected from the lidar between 5 and 7 km altitude ASL, in the morning of 24th October (Figure 6). However, the ceiling heights of the UAS flights that day were limited to 4 km, owing to battery limitations, thereby preventing the observation of the dust layers. Throughout the first dust event, UAS flights intersected the elevated dust layer, as confirmed by lidar observations, on four occasions: 25th, 27th, 29th, and 31st of October. These instances coincided with air masses originating from NA. As shown in Figure 6 and Table 1, during the second dust event, the elevated dust layers ranged between 1.4 and 3.8 km altitude ASL, all of which were effectively captured by the UAS observations.

Figure 8 shows the 120-hour backward trajectories of air masses arriving at the elevated dust layers observed over Orounda, corresponding to the specific UAS flight times noted in Table 2.1. The trajectories associated with the first dust event indicate that these air parcels originated from dust-source regions in North and Central Africa. Subsequently, they traversed the Mediterranean and arrived in Cyprus in a span of 3 days. The trajectories linked to the second dust event reveal that these air parcels originated from dust sources within the ME basin, particularly from counties such as Iraq, Syria, and Jordan. It appears

295

**Figure 8.** The 120-h HYSPLIT backward trajectories starting over Orounda from the elevated dust layers observed at the ToT of the UAS and shown in Table 1 during the Cyprus Fall Campaign 2021. Colour coding refers to the height of the trajectories. The dates in the plot depict the arrival date of the air mass over the island at the arrival height corresponding to the dates given in Table 1

that these air masses were initially uplifted from near ground level and then advected, reaching Cyprus at altitudes ranging between 1.5 and 2.8 km ASL.

Figure 9 shows the vertical structure of mass concentrations derived from combined OPC observations during the campaign. The profiles are characterised by similar structure as the lidar overview (Figure 6). During 18th and 24th of October, calculated mass concentrations remained below  $50\,\mu\mathrm{g/m^3}$  along the flying height column. On 25th of October, mass concentration increased up to  $150\,\mu\mathrm{g/m^3}$  from ground up to  $2\,\mathrm{km}$  ASL, extending to  $3\,\mathrm{km}$  ASL by 27th of October.

On 27th of October 2021, two layers of mass concentration up to  $150 \,\mu\text{g/m}^3$  were observed. The first layer was found at 0.5 km ASL, and the second between 2.5 and 3 km ASL, which was dominated by coarse-mode particles larger than 3.4  $\mu$ m.

On 28th of October, an aerosol layer of particles smaller than 3.4  $\mu m$  and mass up to  $40 \,\mu g/m^3$  was found between 3 and 3.5 km ASL, whereas this layer was gradually descended to lower altitudes along with an increase on the concentration of small particles up to  $80 \,\mu g/m^3$  in the following days. On 4th of November, the atmosphere was aerosol-free, with low concentration of particles smaller than 3.4  $\mu m$ , and almost no particles larger than 3.4  $\mu m$ .

From 14th to 18th of November, the mass concentrations of particles smaller than  $3.4\,\mu\mathrm{m}$  was ranging between 20 and  $50\,\mu\mathrm{g/m^3}$  from ground up to  $3{,}000\,\mathrm{m}$ , while mass concentration of coarser particles reached up to  $180\,\mu\mathrm{g/m^3}$ . Overall, the second event was characterised with higher concentration of coarse-mode particles than the first dust event.

**Figure 9.** Mass concentration profiles derived from UAS-OPC observations during the campaign period. Asterisks (\*) denote profiles calculated exclusively from UCASS observations, as no POPS measurements were conducted on those days.

Figure 10 shows the particle number (a) and volume (b) size distributions within elevated dust layers referenced in Table 2.1, measured during the campaign. The layers characterized by Relative Humidity exceeding 90% were intentionally excluded from these computations to minimize the influence of high relative humidity and cloud presence on the dust layers.

Overall, the particle number size distributions exhibit similar patterns, but display distinct regional characteristics. A clear separation between fine and coarse particles is evident around  $0.7 \, \mu m$ . The NA dust cases are characterized by lower concentrations in the fine fraction and a well-defined coarse-mode peak between approximately 1.5 and  $3.0 \, \mu m$ . In contrast, the ME dust cases display higher fine-mode concentrations and a broader, more extended coarse mode reaching up to about  $10 \, \mu m$ .

In cases of NA dust, a prominent peak is often observed in the coarse mode, typically between 1.5 and  $3.0\,\mu m$  in diameter. Conversely, in cases associated with dust episodes from the ME, a consistent feature is the presence of a minimum value in the volume size distribution at  $0.7\,\mu m$ . This specific value is indicative of a clear differentiation between the two modes, suggesting distinct fine and coarse aerosol components.

In the volume size distributions, the ME cases also exhibit a distinct minimum near  $0.7 \,\mu\text{m}$ , marking the transition between fine and coarse dust particles. This minimum indicates the presence of two clearly separated dust size regimes, reflecting differences in source characteristics and transport history. The broader coarse-mode distribution in the ME dust indicates a wider range of particle sizes and potentially greater variability in source characteristics, whereas the narrower and more sharply peaked coarse mode in the NA dust points to a more uniform particle population and different source or transport influences. Some ME cases show an upward trend in the coarse tail up to  $20 \,\mu\text{m}$ , potentially indicating contributions from large, near-source dust particles with minimal atmospheric processing (e.g., Ryder et al. (2013); Weinzierl et al. (2017); Denjean et al. (2016)), a feature less evident in the more NA cases.

330

**Figure 10.** Particle number and volume size distribution within the elevated dust layers shown in Table 2.1, as calculated by OPC flights during the campaign period. The blue lines represent measurements from the first dust event from NA, and red lines from the second dust event from ME.

Figure 11 shows the average particle number and volume size distributions along with the lognormal fitting for the two regions, NA and ME, based on all cases presented in Figure 10. The normalized root mean square error (NRMSE) for the lognormal fitting of the number size distribution (dN/dlogD) was 3.8% for the ME case and 4.3% for the NA case, indicating a reasonably good fit in both cases. The NRMSE for the lognormal fitting of the volume size distribution (dV/dlogD) was 10% for the ME case and 13% for the NA case.

In episodes originating from NA, the volume size distribution exhibits a distinct fine-mode peak with a geometric mean diameter (GMD) of 0.23,µm and a geometric standard deviation (GSD) of 1.44, while the coarse mode shows a more pronounced peak at a GMD of 2.2,µm with a GSD of 1.9. For cases originating from the ME, a fine-mode peak is evident at 0.25 µm with a GSD at 1.5, accompanied by a broader peak in the coarse-mode with a GMD of 4.8 µm and a GSD of 2.5. This difference in size distribution reflects the differing aerosol composition and sources between the two regions. The pronounced peak in the fine-mode observed during ME cases indicates a higher concentration of smaller aerosols. Conversely, the broad coarse-mode peak suggests a substantial contribution from larger particles, which may arise from differences in emission sources and atmospheric transport processes relative to NA. Nevertheless, the present analysis does not allow for a clear attribution of these effects or their relative importance. These findings underscore the complexity of aerosol characteristics in the region and the varied sources driving particle emissions (Pikridas et al., 2018).

Particles smaller than 1 µm exhibit concentrations approximately one order of magnitude higher in the ME relative to NA. This observation aligns with findings of Christodoulou et al. (2023); Bimenyimana et al. (2025), which highlight the substantial contribution of ME emissions, with peak diameter between 0.18 and 0.35 µm. The coarse-mode of the particle volume size distributions observed over Cyprus are broadly consistent with previous measurements from major desert dust field campaigns. During ME events, the coarse-mode peaks around 4.8 µm and extends up to approximately 10 µm, indicating the presence of larger particles and a broad size range. This is comparable to source-region observations from campaigns such as Fennec and SAMUM-1, where coarse-mode peaks typically occur between 5–8 µm (Ryder et al., 2018; Weinzierl et al., 2009). In contrast, dust originating from NA exhibits a sharper coarse-mode peak near 2.2 µm, consistent with size distributions observed in transported dust during campaigns such as SAMUM-2, AER-D, and SALTRACE, where peaks were found between 3–4 µm due to gravitational settling during long-range transport (Weinzierl et al., 2017; Ryder et al., 2019; Formenti et al., 2011). These results suggest that the broader coarse-mode volume size distributions observed in ME cases reflects the influence of more proximal or freshly resuspended dust sources, while the narrower NA distributions represent more aged, atmospherically processed dust arriving from distant Saharan sources.

**Figure 11.** Particle number (left panel) and volume (right panel) size distribution within the elevated dust layers shown in Table 2.1, as calculated by OPC flights during the Cyprus Fall Campaign. The data are averaged and classified based on their origin, distinguishing between cases originating from NA and the ME. A lognormal distribution is fitted to the data, shown in dashed line.

# 3.3 Airborne observations - Impactors

In addition to the height-resolved measurements acquired through UAS-based OPCs, impactors were deployed to collect airborne dust samples across different altitudes. Subsequently, these samples were analyzed using SEM, providing complementary information on the morphological and chemical characteristics of the collected particles. This dual approach improves the overall understanding of the vertical dust distribution and its composition, contributing valuable insights into atmospheric aerosol dynamics. Table 2 provides details on the altitude range of the collected particles and their origin source as revealed by the back-trajectories. A total of five samples were collected during the first dust event, which originated from NA, and eleven samples were obtained during the second event, from the ME.

Figure 12 shows the mean relative abundance of aerosol types as a function of particle size for the ME and NA sample sets. In both regions, mineral dust dominates the composition, accounting for more than 80% of total particle abundance across most size ranges. The relative contribution of dust slightly decreases to about 70% in one coarse-mode bin for each region. Sulfate-and sea-salt-rich particles contribute modestly, primarily in the submicron and fine-mode size ranges (below 2,μm). The NA samples exhibit a slightly higher proportion of sulfate compared to ME samples. "Other" particles are mainly mixtures of the three dominant types but also include non-classified substances. Conversely, the ME samples show a marginally larger fraction of coarse "other" particles at diameters exceeding 10,μm. Overall, both datasets confirm that mineral dust is the dominant aerosol type, with subtle regional differences.

**Table 2.** Details on the GPaC samples collected on-board the UAS, including sampling ID name, date, altitude range where the impactor was open, sampling period, UAS average airspeed during the sampling duration, total number of particles that were analysed under SEM, and origin region of the sampling layers based on HYSPLIT back-trajectory analysis.

| Commle ID    | Date       | Height ASL | Duration | Airspeed | Total particles | Origin |
|--------------|------------|------------|----------|----------|-----------------|--------|
| Sample ID    |            | (km)       | (sec)    | (m/s)    | analysed        | Region |
| CYI-GPAC-405 | 27/10/2021 | 2.4-2.9    | 138      | 11.6     | 838             | Africa |
| CYI-GPAC-407 | 27/10/2021 | 1.5-2.2    | 221      | 12.5     | 571             | Africa |
| CYI-GPAC-408 | 27/10/2021 | 2.6-3.3    | 121      | 11       | 787             | Africa |
| CYI-GPAC-411 | 28/10/2021 | 2.4-3.5    | 280      | 12.1     | 1156            | Africa |
| CYI-GPAC-412 | 29/10/2021 | 1.9-3.6    | 411      | 11.7     | 514             | Africa |
| CYI-GPAC-501 | 29/10/2021 | 0.7-1.7    | 400      | 11.5     | 560             | Africa |
| CYI-GPAC-601 | 13/11/2021 | 1.6-2.2    | 134      | 12       | 503             | M.East |
| CYI-GPAC-602 | 13/11/2021 | 0.7-1.3    | 188      | 12       | 1003            | M.East |
| CYI-GPAC-603 | 14/11/2021 | 1.9-2.3    | 100      | 11.8     | 898             | M.East |
| CYI-GPAC-604 | 14/11/2021 | 1.4-1.8    | 72       | 11.8     | 1280            | M.East |
| CYI-GPAC-605 | 14/11/2021 | 1.72-1.74  | 247      | 11.3     | 797             | M.East |
| CYI-GPAC-607 | 15/11/2021 | 1.8-4.3    | 630      | 12.3     | 2027            | M.East |
| CYI-GPAC-609 | 15/11/2021 | 1.72-1.74  | 243      | 11       | 1015            | M.East |
| CYI-GPAC-610 | 16/11/2021 | 1.33-1.83  | 99       | 12.2     | 935             | M.East |
| CYI-GPAC-611 | 16/11/2021 | 2.14-2.63  | 97       | 12.2     | 577             | M.East |
| CYI-GPAC-612 | 16/11/2021 | 1.93-1.950 | 246      | 12.6     | 777             | M.East |
| CYI-GPAC-701 | 18/11/2021 | 0.86-2.74  | 412      | 12.3     | 1508            | M.East |

Figure 12. Relative abundance of aerosol of particles collected on GPaC as averaged for ME (left) and NA (right) samples. The relative abundance (y-axis) denotes the fraction of particles in each sample that are dominated by the indicated composition.

Figure 13 shows the particle number size distribution averaged and classified based on their origin, distinguished between cases originating from NA and the ME. The comparison shows that the ME sample set consistently contains more particles than the NA set, especially in the fine particle range. From submicron to  $3\,\mu\mathrm{m}$  diameters, ME concentrations are about 2–3 times higher than NA, with the largest enhancement near  $0.8\,\mu\mathrm{m}$ . Beyond  $10\,\mu\mathrm{m}$ , the difference diminishes, and the two distributions converge. This indicates that ME primarily enhances the smaller particle population, while coarse particles remain similar between the two sets.

The GPAC (Figure 13) and OPC (Figure 11) number size distributions show overall good agreement, despite differences in technique and resolution. However, the GPAC derived size distribution lacks sufficient resolution below  $0.42\,\mu\mathrm{m}$  to confirm the fine-mode peak observed in the OPC data. GPAC tends to report slightly higher concentrations from the submicron to  $2\,\mu\mathrm{m}$  diameters, but at larger diameters (>5–10  $\mu\mathrm{m}$ ) the slope and magnitude align closely between the two datasets. Importantly, both instruments consistently distinguish regional contrasts, with the ME exhibiting higher number concentrations than NA across the size spectrum. This strong cross-instrument consistency, particularly in the coarse-mode particles, reinforces confidence in the reliability of the measurements and demonstrates that GPAC and OPC provide complementary and robust insights into the particle number size distributions.

**Figure 13.** Averaged particle number size distribution within the elevated dust layers shown in Table 2.1, as calculated by the GPAC samples during the Cyprus Fall Campaign. The data are averaged and classified based on their origin, distinguishing between cases originating from NA and the ME.

Figure 14 shows ternary plots of the mean single-particle elemental composition. The Cyprus cases can be differentiated from the other source regions in the Ca-Al-Mg and K-Ca-Fe ternary plots. In the Ca-Al-Mg ternary diagram, dust of ME origin observed over Cyprus exhibits a distinct shift toward the Ca-rich vertex, whereas NA samples align more closely with previously reported Saharan measurements (Kandler et al., 2020), where the same analytical procedure was applied. This pattern likely reflects intrinsic differences in the mineralogical composition and soil structure of the respective source regions. A similar trend is observed in the K-Fe-Ca ternary diagram, where Ca enrichment again serves as a distinguishing feature between the sources; however, the Cyprus samples exhibit relatively higher Fe and lower K contents compared to previously reported Saharan dust. Overall, the Cyprus dust samples show slightly lower inter-particle variability than those from other regions, as indicated by the reduced extent of the statistical confidence envelope.

Figure 14c shows the relative abundance of particles with elemental compositions resembling common clay minerals. Distinct regional fingerprints are evident, where samples originating from NA exhibit mineralogical patterns closely matching those observed for Saharan dust collected on Tenerife, whereas the ME samples do not correspond clearly to any previously defined compositional group. The NA samples are characterized by elevated proportions of kaolinite-like minerals and calcium-rich phases, consistent with more weathered, carbonate-bearing source soils typical of NA dust (Rodríguez-Navarro et al., 2018; Kandler et al., 2020). In contrast, the ME samples display signatures dominated by illite and muscovite, indicative of less weathered, aluminosilicate-rich material. These compositional differences provide a clear mineralogical distinction be-

tween the two dust source regions and underscore the potential of single-particle elemental analyses to trace the provenance of airborne mineral dust reaching Cyprus.

Figure 14. (a, b) Ternary diagrams showing the mean single-particle atomic composition of the dust fraction from different samples collected during the Cyprus Fall Campaign (green symbols), along with data from previous campaigns reported by Kandler et al. (2020). Shaded regions denote bootstrapped 95% confidence intervals of the mean composition. The bottom panel illustrates the mean number contribution of particles resembling the indicated clay mineral classes; color coding and confidence intervals are consistent with panel (a).

## 395 4 Summary and Conclusions

This study demonstrate the effectiveness of a novel, cost-efficient methodology for quantitative characterization of airborne dust particles using a sensor package that integrates OPCs and impactors deployed on UAS. This approach enables high-resolution vertical profiling and robust detection of coarse particles within elevated dust layers. Such measurements are essential for improving the reliability of satellite observations and air quality assessments. Beyond refining remote-sensing retrievals, these results enhance understanding of how regional dust transport influences weather patterns, visibility, and climate, while showcasing the value of small, flexible aerial system for atmospheric research.

UAS-based measurements and remote-sensing observations were performed between 18th of October and 18th of November 2021, in the Nicosia basin in Cyprus. Two distinct dust episodes were observed, originating from NA and the ME, respectively. Each event exhibited a unique signature in terms of layer altitude, particle size distribution, and chemical composition. Together,

they offer a valuable opportunity to investigate the atmospheric transport mechanisms and physical—chemical properties of dust aerosols in the region.

Throughout the campaign, a total of 36 UAS atmospheric flights were conducted, in conjunction with ground-based remote-sensing observations. The mean daily-averaged AOD measured by local sun-photometers during the NA dust event was approximately 0.22, compared to about 0.27 for the ME event. A 120-hour HYSPLIT back-trajectory analysis conducted during the Cyprus Fall Campaign shows that the air masses associated with dust events originate from two primary regions. During Saharan dust intrusions, the trajectories trace back to NA, -specifically Algeria, Libya, Egypt, and Mauritania- while during ME dust outbreaks, the sources are primarily Syria, Iraq, Jordan, and Saudi Arabia. Dust layers originating from NA were observed at altitudes that reached up to 7 km, while those transported from the ME typically reached lower altitudes, with maxima around 3.8 km.

NA dust episodes exhibit lower fine-mode concentrations compared to ME cases, although both show volume size distribution peaks around 0.25 µm. In coarse mode, NA cases exhibit a sharply peaked distribution with a maximum near 2.2 µm, whereas ME cases display a broader distribution peaking around 4.8 µm and extending up to 10 µm. Notably, the ME coarse-mode PSD is extremely broad, while the NA distribution declines rapidly beyond its peak. The GPaC collection efficiency reaches nearly 1 for particles between 4 and 14 µm, with lower efficiency outside this range (around 50%), highlighting the promising capability of the GPaC method across the particle size spectrum compared to the UCASS.

Variations in particle size distribution are attributed to differences in the mineralogy and chemical composition of dust originating from different source regions. In this study, the combined OPC and GPAC approach deployed on UASs proved highly effective for identifying mode-specific composition and aerosol type. The GPAC measurements enabled verification of the particle composition associated with the observed OPC bi-modal particle size distribution—most evident in the ME samples—demonstrating that both fine- and coarse-mode particles were predominantly composed of dust.

Overall, this study reveals two distinct compositional fingerprints for Cyprus dust. NA-sourced samples show higher kaolinite-like and Ca-rich signatures, while ME sources are shifted toward illite/muscovite- and Fe-enriched compositions. This separation highlights Cyprus as a unique receptor where contrasting mineralogical regimes converge, allowing clear discrimination of source-dependent microphysical and chemical properties.

*Data availability.* All data can be provided by the corresponding authors upon request. Data can also be found in Zenodo at https://doi.org/10.5281/zenodo.17723607.

# Appendix A: Sampling with GPaC

**Table A1.** Classification criteria value ranges in terms of elemental ratios for classifying particles into a certain class. A) for clay-mineral like groups. B) for distinguishing dust from other particles. Note that |X| = X / (Na+Mg+Al+Si+P+S+Cl+K+Ca+Ti+Cr+Mn+Fe), the element symbol representing the atomic (molar) concentration of that element. Due to its occurrence in dust as well as in sea-salt, Na is not regarded for the dust disambiguation. All conditions have to be met for a positive classification. Unmatched particles are classified as 'other'.

| (A)                 | Kaolinite-like | Illite-/Muscovite-like | Chlorite-like |
|---------------------|----------------|------------------------|---------------|
| Na+Cl+2 SI /  Al+Si | < 0.25         | < 0.25                 | < 0.25        |
| Al+Si               | >0.7           |                        |               |
| K+Al+Si             |                | >0.7                   |               |
| Mg+Fe+Al+Si         |                |                        | >0.7          |
| Na/(Al+Si)          | <0.1           | <0.2                   | <0.1          |
| Mg/(Al+Si)          | <0.2           | <0.2                   | 0.25 - 0.8    |
| Al/(Al+Si)          | 0.444 - 0.545  | 0.31 – 0.6             | 0.333 - 0.6   |
| K/(Al+Si)           | <0.1           | 0.1 – 1                | <0.1          |
| Ca/(Al+Si)          | <0.1           | <0.2                   | <0.3          |
| Fe/(Al+Si)          | <0.2           | <0.2                   | 0.2 - 1       |

| <b>(B)</b> |                                                                 |          |
|------------|-----------------------------------------------------------------|----------|
| Dust       | Mg+Si+Al+K+Ca+Ti+Fe  /  Mg+Al+Si+P+S+Cl+K+Ca+Ti+Cr+Mn+Fe  > 0.7 | S  > 0.3 |
| Sea-salt   | Na+Mg+Cl  > 0.7                                                 | S  < 0.3 |
| Sulfate    |                                                                 | S  > 0.3 |

**A1** 

Author contributions. MK: Conceptualization, participation in field campaigns and data acquisition, data analysis and interpretation, writing—original draft; FM: Conceptualization, supervision, participation in field campaigns and data acquisition, theoretical background, placing the subject matter in the wider context; AP: Lidar observations in Nicosia, participation in field campaigns and data acquisition.; KK: SEM analysis on samples; CLR, FM, JS: Critical review of scientific content; AL, CK: Coordination of UAS flights; CS: Provision of calibration data of the UCASS; TT: Provision of technical support for the POPS; REM: Lidar observations in Limassol; All authors have contributed by writing, reviewing and editing.

Competing interests. The authors declare no conflicts of interest.

Acknowledgements. The CARO-LIM NF operation is supported by the 'EXCELSIOR' project funded by the European Union's Horizon 2020 research and innovation programme under Grant Agreement No 857510, from the Government of the Republic of Cyprus through the Directorate General for the European Programmes, Coordination and Development and the Cyprus University of Technology. REM also acknowledge the ATARRI project funded by the European Union's Horizon Europe Twinning Call (HORIZON-WIDERA-2023-ACCESS-02) under the grant agreement No 101160258. The EMME-CARE project has received funding from the European Union's Horizon 2020 research and innovation programme under grant agreement No. 856612 and the Cyprus Government.

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
