# Peer review of "Microphysical and Compositional Differences Between Saharan and Middle Eastern Dust Revealed by UAS Observations"

_EGUsphere, 2025_

## Referee Comment (RC1)

Review of a manuscript titled

**"Microphysical and Compositional Differences Between Saharan and Middle Eastern Dust Revealed by UAS Observations"**

**General comments:**

The manuscript presents an interesting and timely study on airborne dust characterization using a low-cost sensor package integrating optical particle counters (OPCs) and impactors on UAS platforms. The combination of in situ size distributions with offline SEM analysis of sampled particles is valuable, and the focus on Saharan and Middle East dust outbreaks is relevant for both regional air quality and long-range transport studies.

However, the manuscript has two major weaknesses that, in my view, must be addressed before the results can be considered robust:

1. The calibration and sizing of the OPCs are not treated with sufficient rigor, especially given that the instruments were originally calibrated for different particle types than mineral dust.
2. The claim of providing vertical information on dust is not fully supported by the way the vertical structure is presented and analyzed.

Below, I detail these points and offer suggestions for improvement.

**1. Calibration and sizing of the OPCs in Section 2.3**

The study relies heavily on OPC-derived particle number and volume size distributions to infer dust loadings and to derive mass concentrations. However, the description and justification of the calibration procedure for the OPCs, as well as their suitability for mineral dust, are not adequately presented.

- The OPCs are inherently sensitive to particle refractive index, shape, and the instrument's optical configuration (wavelength, angular range). In the present work, it appears that the instruments were used with standard calibrations (e.g. PSL or other reference aerosols) and then directly applied to Saharan and Middle East dust without a systematic correction for the optical properties of dust.

- Mineral dust is non-spherical and has a refractive index and absorption that differ significantly from calibration aerosols such as PSL. This implies that the "optical equivalent diameter" reported by the OPCs can differ substantially from the true geometric or aerodynamic diameter. This is especially critical for coarse-mode particles, which are central to the paper's conclusions.

- The manuscript uses OPC data to compute volume size distributions and dust mass concentrations based on assumed densities (e.g. ~2.6 g cm⁻³ for coarse mineral dust), but the uncertainties introduced by the optical sizing are not quantified. In practice, the combination of refractive index mismatch and non-sphericity can lead to systematic size biases that propagate non-linearly into volume and mass.

**Suggestions:**

- Provide a detailed description of the calibration of each OPC used, including:

    - Calibration aerosol (material, refractive index).

    - Calibration method (Mie model, reference instruments, etc.).

    - Operational size range where the calibration is considered reliable.

- Discuss explicitly how the refractive index and non-sphericity of mineral dust affect the OPC response. At minimum, this should be treated in an uncertainty analysis, and key conclusions should be qualified accordingly.

- If possible, reprocess the OPC data using an assumed refractive index representative of Saharan/Middle East dust, and appropriate Mie calculations for each instrument's wavelength and collection angles. This would help bring the sizing onto a more physically consistent basis.

- Clearly distinguish throughout the text between "optical equivalent diameter" and other size metrics (geometric, aerodynamic) and avoid implying that the OPC diameters are exact physical sizes.

- Where mass concentrations are derived from OPC volume and assumed densities, include an estimate of the uncertainty due to optical sizing (not only due to density assumptions). A simple sensitivity analysis using a range of plausible dust refractive indices would already improve confidence in the results.

Given that the study's conclusions rely on differences in coarse-mode size distributions between NA and ME dust, the robustness of these conclusions depends strongly on the soundness of the OPC calibration and size retrieval.
* * *
**2. Lack of truly vertical-resolved dust information**

The manuscript emphasizes the capability of UAS-based OPCs and impactors to provide high-resolution vertical profiling of dust layers. However, the presentation of vertical structure is

relatively limited, and the reader is left without a clear, quantitative view of the vertical distribution of dust properties.

- While flights are described as targeting specific altitude ranges and back-trajectory analysis is used to differentiate NA and ME sources, the figures and discussion focus largely on layer-averaged or campaign-averaged size distributions (e.g. Fig. 11 showing number and volume size distributions within "elevated dust layers").

- The vertical variability within these layers is only briefly mentioned (e.g. via standard deviations over entire flights), but the manuscript does not provide systematic vertical profiles of particle number, volume, or derived mass concentration as a function of altitude.

- The impactor/SEM analysis adds valuable information on morphology and composition, but it is presented mainly by layer or by origin type, rather than as vertical cross-sections or profiles that demonstrate how size, composition, or mineralogy change with height.

**Suggestions:**

- Include explicit vertical profiles of key quantities measured by the OPCs:

    - Particle number concentration in relevant size bins.

    - Volume (or mass) concentration profiles for fine and coarse modes.

    - Where appropriate, size-resolved profiles (e.g. contour plots of number or volume vs altitude and diameter).

- Clarify how "elevated dust layers" are defined (from lidar, back-trajectories, or OPC signal) and how the altitude bounds listed in Table 2 relate to the vertical profiles.

- For the impactor data, consider summarizing composition/morphology as a function of altitude (e.g. showing how the relative contributions of different mineralogical classes or elemental ratios vary with height within a given event).

- The manuscript currently states that the approach enables "high-resolution vertical profiling" and "robust detection of coarse particles within elevated dust layers"; these statements would be more convincing if supported by concrete vertical cross-sections or profiles from representative flights.

- If vertical resolution is constrained by flight logistics (e.g. step-wise levels rather than continuous profiling), this limitation should be clearly stated, and the implications for resolving fine vertical structures (such as multiple dust layers or sharp layer boundaries) should be discussed.

Overall, the paper would benefit from a much clearer and more quantitative depiction of the vertical structure, beyond layer-averaged statistics.

**Specific comments:**

Abstract: The back trajectory information in lines 8-9 doesn't represent the sampling altitudes of the UAS, but misleads the readers when presented along with the UAS dust information.

Introduction: This study didn't achieve the two objectives outlined in lines 61 -63.

Line 85: Is this AOD based on the height of the lidar column or the UAS column or natural atmospheric column? How does it relate to the UAS measurement and lidar retrieval? What is the uncertainty range?

Line 87, What "height-resolved information" does this study provide? Please clarify.

Section 2.2: The sampling periods are limited. How does this limitation provide representative information for the sampling day? What are the ascending and descending rates of the flights? Does the inlet operate under isokinetic conditions during the flight?

Line 163: What is this size? Aerodynamic size? If so, how does it relate to the optical size measured by POPS and UCASS?

Line 177-178: One sample per flight? If so, no vertical resolution of your samples.